# Local audit of empiric antibiotic therapy in bacteremia: A retrospective cohort study

**Anthony D. Bai**[1]*, **Neal Irfan**[2], **Cheryl Main**[3], **Philippe El-Helou**[1], **Dominik Mertz**[1]

**1** Division of Infectious Diseases, McMaster University, Hamilton, Ontario, Canada, **2** Division of Medicine, McMaster University, Hamilton, Ontario, Canada, **3** Division of Medical Microbiology, Department of Pathology and Molecular Medicine, McMaster University, Hamilton, Ontario, Canada

\* anthony.bai@medportal.ca

**Data Availability Statement:** All relevant data are within the manuscript and its supporting information files.

**Funding:** The authors received no specific funding for this work.

## Abstract

### Background

It is unclear if a local audit would be useful in providing guidance on how to improve local practice of empiric antibiotic therapy. We performed an audit of antibiotic therapy in bacteremia to evaluate the proportion and risk factors for inadequate empiric antibiotic coverage.

### Methods

This retrospective cohort study included patients with positive blood cultures across 3 hospitals in Hamilton, Ontario, Canada during October of 2019. Antibiotic therapy was considered empiric if it was administered within 24 hours after blood culture collection. Adequate coverage was defined as when the isolate from blood culture was tested to be susceptible to the empiric antibiotic. A multivariable logistic regression model was used to predict inadequate empiric coverage. Diagnostic accuracy of a clinical pathway based on patient risk factors was compared to clinician's decision in predicting which bacteria to empirically cover.

### Results

Of 201 bacteremia cases, empiric coverage was inadequate in 56 (27.9%) cases. Risk factors for inadequate empiric coverage included unknown source at initiation of antibiotic therapy (adjusted odds ratio (aOR) of 2.76 95% CI 1.27–6.01, P = 0.010) and prior antibiotic therapy within 90 days (aOR of 2.46 95% CI 1.30–4.74, P = 0.006). A clinical pathway that considered community-associated infection as low risk for *Pseudomonas* was better at ruling out *Pseudomonas* bacteremia with a negative likelihood ratio of 0.17 (95% CI 0.03–1.10) compared to clinician's decision with negative likelihood ratio of 0.34 (95% CI 0.10–1.22).

### Conclusions

An audit of antibiotic therapy in bacteremia is feasible and may provide useful feedback on how to locally improve empiric antibiotic therapy.

**Competing interests:** The authors have declared that no competing interests exist.

## Introduction

Bloodstream infections are associated with significant mortality and morbidity [1]. Empiric antibiotic therapy is an important component in the treatment of bacteremia. Inadequate empiric antibiotic coverage or a delay in getting the adequate antibiotic therapy has been associated with a higher mortality [2, 3]. However, indiscriminate use of broad-spectrum antibiotics contributes to antibiotic resistance [4]. Therefore, clinicians must balance between ensuring adequate empiric coverage and minimizing the unnecessary use of broad-spectrum antibiotics.

Clinicians make decisions on empiric antibiotics on their own early in the disease course. Antimicrobial stewardship interventions such as reassessment of antibiotic therapy are often done on day 3 of antibiotic therapy [5, 6]. As a result, optimization of empiric antibiotic therapy is often a missed opportunity. A challenge in optimizing empiric antibiotic therapy is that it is unclear what are the circumstances in which clinicians choose inadequate empiric coverage. Another unknown is whether a clinician's decision is better than a simple clinical pathway based on specific patient risk factors in predicting what to cover empirically.

Prior studies on empiric antibiotic coverage are very heterogeneous in terms of determinants and proportion of patients treated with inadequate empiric coverage [7]. Heterogeneity may be related to differences in antibiotic resistance and clinician prescribing patterns across various hospitals [7]. This heterogeneity makes it difficult to apply these study findings to local hospital settings. Therefore, local data may be the best way to drive changes in empiric antibiotic prescribing pattern.

We performed a local audit of empiric antibiotic therapy in patients with bacteremia across 3 hospitals in Hamilton, Ontario, Canada to evaluate the usefulness of an audit in providing feedback on how to improve local empiric antibiotic practices. We had three specific objectives: 1) assess the proportion of cases on inadequate empiric coverage; 2) determine risk factors for inadequate empiric coverage; 3) compare the accuracy of a simple clinical pathway based on patient risk factors to clinician's decision in predicting when to empirically cover for methicillin-resistant *Staphylococcus aureus* (MRSA) or *Pseudomonas aeruginosa* bacteremia.

## Methods

### Study design

We conducted a retrospective cohort study at three acute care academic hospitals in Hamilton, Ontario, Canada. The Hamilton Integrated Research Ethics Board approved this study and waived the requirement for individual patient consent.

### Patient selection

During the month of October in 2019, all consecutive positive blood culture results from patients in these three acute care hospitals were extracted from the electronic medical record. Patients could be an outpatient seen in the emergency department or an inpatient admitted to the hospital. A patient could be included in the study more than once if they had different blood cultures growing different organisms at different times.

Bacteremia events were excluded if they had any of the following:

1. Monomicrobial growth of an organism that is likely to be a contaminant including coagulase negative *Staphylococcus* (other than *S. lugdunensis*), *Bacillus* (other than *B. anthracis*), *Paenibacillus*, *Corynebacterium* (other than *C. jeikeium*), *Cutibacterium acnes* and *Micrococcus* species [8]

2. Growth of *Candida* species

3. The patient was not expected to survive and was palliated within 72 hours of blood culture collection

## Data collection

Patient medical information was all recorded in real time on the same day of the patient encounter. Data was extracted from the patient electronic chart system during the time period from October 14, 2020 to Dec 7, 2020. The following information was collected: demographics, hospital site, admitting service, comorbidity as per the Charlson comorbidity index [9], setting of how the infection was acquired, suspected infectious source when empiric antibiotic therapy was started based on documentation on patient chart, infectious source as per the physician on hospital discharge, risk factors for antibiotic resistant organisms, severity of infection as per the Pitt bacteremia score within the first 24 hours of blood culture collection [10], blood culture timing and results, antibiotic therapy, and death at 30 days. The data was de-identified after it was entered into the study database.

## Definition of variables

The infections were classified as community, hospital or healthcare associated based on where they were acquired using established definitions [11]. If the positive blood culture was collected after 48 hours of intensive care unit (ICU) stay, then it was defined as ICU associated.

Persistent bacteremia was defined as a positive blood culture growing the same organism taken >24 hours after the initial positive culture while on adequate antibiotic therapy. Recurrent bacteremia was defined as a positive blood culture growing the same organism as the first positive blood culture after completion of adequate antibiotic therapy.

Prior cultures of antibiotic resistant organisms were defined as being within the time period of 12 months prior to the current bacteremia episode.

## Hospital policy

A clinician that took care of the patient independently decided on empiric antibiotic therapy. Clinicians had access to hospital guideline recommendations on empiric antibiotic therapy for specific infection syndromes (Table 1) as well as site-specific antibiogram.

## Adequate empiric coverage

Empiric antibiotic therapy was defined as antibiotics that the patient received within 24 hours after blood culture collection. The time point of 24 hours was chosen, because the majority of bacteremia would have positive blood culture results by this time, allowing physicians to change to more targeted antibiotic therapy [12]. As well, the cut-off of 1 day to define empiric antibiotic therapy has been used previously [13].

Empiric coverage was considered adequate if the bacterial isolate from the blood culture was shown to be susceptible to the antibiotic used. For polymicrobial bacteremia, antibiotic therapy was considered adequate if the antibiotic therapy covered all of the organisms grown in the blood culture.

At the microbiology lab, matrix assisted laser desorption ionization-time of flight mass spectrometry was used to identify the bacteria species. Susceptibility testing was done at the microbiology lab based on clinical laboratory standards institute (CLSI) guidelines. The antibiotic also needed to be an appropriate dose for treating bacteremia. Oral antibiotics needed to

**Table 1. Hospital guidelines on empiric antibiotic therapy.**

| Syndrome | Recommended empiric antibiotic therapy |
|---|---|
| Community associated meningitis | All of the following: |
| | Vancomycin IV |
| | Ceftriaxone IV |
| | Ampicillin IV if risk factors for *Listeria* |
| Post neurosurgical meningitis CSF shunt infection | All of the following: |
| | Vancomycin IV |
| | Ceftazidime IV or Meropenem IV |
| Community associated pneumonia | Any of the following: |
| | Ceftriaxone IV plus Azithromycin IV or PO |
| | Levofloxacin IV or PO |
| | Moxifloxacin IV or PO |
| Hospital associated pneumonia | Any of the following: |
| | Piperacillin-Tazobactam IV |
| | Meropenem IV |
| | If known MRSA colonization, add Vancomycin IV |
| Community associated intra-abdominal infection | Ceftriaxone IV plus Metronidazole IV or PO |
| | If APACHE 2 score > = 15 or healthcare associated infection, then Piperacillin-Tazobactam IV |
| | If beta-lactam allergy, then use Ciprofloxacin IV or PO plus Metronidazole IV or PO instead |
| Hospital associated intra-abdominal infection | Piperacillin-Tazobactam IV |
| | If known MRSA colonization, add Vancomycin IV |
| | If colonization or history of ESBL or SPICE organisms, then use Meropenem IV instead |
| Community associated urinary tract infection | Any of the following: |
| | Ceftriaxone IV |
| | Tobramycin IV |
| Uncomplicated cellulitis | Cefazolin IV |
| | If beta-lactam allergy, then use Vancomycin IV or Clindamycin IV instead |
| | If MRSA risk factors, then use Vancomycin IV instead |
| Necrotizing fasciitis | All of the following: |
| | Piperacillin-Tazobactam IV |
| | Clindamycin IV |
| | If MRSA risk factors, then add Vancomycin IV |
| Febrile neutropenia | Any of the following: |
| | Piperacillin-Tazobactam IV |
| | Ceftazidime IV |
| | Ciprofloxacin IV plus Gentamicin IV |
| | If MRSA colonization or extensive mucositis or septic shock, then add Vancomycin IV |

Positive blood cultures were considered a critical result, so the microbiology lab would phone to notify the clinician who ordered the blood cultures immediately.

have high bioavailability in order to be considered adequate for treating bacteremia (e.g. fluoroquinolones).

Susceptibility testing was not performed in a few cases when an anaerobe was isolated or an organism failed to grow for susceptibility testing. As well, some susceptibility was assumed and

not tested (e.g. penicillin susceptibility for *Streptococcus pyogenes*). For these cases, we referred to the Sanford Guide on Antimicrobial Therapy [14] to determine if the organism was typically susceptible to the antibiotic.

For bacteria with extended-spectrum beta-lactamase (ESBL), we considered Piperacillin-Tazobactam to be inadequate [15]. We defined SPICE organisms to include *Serratia*, *Providencia*, indole positive *Proteus*, *Citrobacter* and *Enterobacter* species. Since these organisms may harbor AmpC beta lactamase that may not be detected in standard susceptibility testing, we considered penicillins and third generation or lower cephalosporins to be inadequate for these organisms.

Unnecessary use of excessively broad-spectrum antibiotics was considered separately from adequate versus inadequate empiric antibiotics. Unnecessary Vancomycin was defined as when Vancomycin was used, but the blood culture and other cultures did not grow MRSA or *Enterococcus* species that were resistant to Ampicillin. Unnecessary *Pseudomonas* coverage was defined as when an empiric antibiotic with activity against *P. aeruginosa* was used, but the blood culture and other cultures did not grow *P. aeruginosa*. Unnecessary use of carbapenem was defined as when a carbapenem was used, but the blood culture and other cultures did not grow an organism with ESBL or a SPICE organism. Note that empiric antibiotic therapy could be both adequate and unnecessary. For example, consider a patient with *Escherichia coli* bacteremia who was treated with empiric Vancomycin and Piperacillin-Tazobactam. The empiric antibiotic therapy was adequate, because the *E. coli* was susceptible to Piperacillin-Tazobactam. However, both the Vancomycin and *Pseudomonas* coverage were unnecessary, because the patient did not have MRSA, *Enterococcus* or *Pseudomonas* bacteremia.

## Statistical analyses

For descriptive analysis, number with percentages was used for categorical variables, and median with interquartile range (IQR) was used for continuous variables. Patients on adequate empiric antibiotic therapy were compared to patients on inadequate empiric antibiotic therapy using Fisher's exact test for categorical variables and Wilcoxon rank-sum test for non-normally distributed continuous variables.

Logistic regression model was used to predict inadequate empiric coverage. Potential predictors were selected for univariate analysis *a priori* based on clinical judgment. Potential predictors included the team that prescribed the empiric antibiotics, setting of how the infection was acquired, unknown infectious source at initiation of antibiotics, prior antibiotics within 90 days and prior surgery within 90 days. All predictors with P <0.2 on univariate analyses were included in the multi-variable model. A forward and backward stepwise regression model based on the Akaike information criterion was used to derive the final multi-variable model of significant predictors.

We compared the diagnostic accuracy of a simple clinical pathway based on a patient risk factor to clinicians' decision regarding MRSA and *Pseudomonas* coverage. The clinicians' decision was the antibiotic coverage decided by the clinician who originally prescribed the empiric antibiotic for that patient. The patient risk factors were selected *a priori* based on clinical judgment. MRSA risk factors included hemodialysis line [16], injection drug use [17], MRSA screening swab [18] and MRSA isolated from prior cultures. *Pseudomonas* risk factors included neutropenia [19], peripherally inserted central catheter (PICC) line [20], *Pseudomonas* isolated from prior cultures [21], prior antibiotics within 90 days [20] and setting of how the infection was acquired [22]. For each risk factor, we calculated the diagnostic properties including sensitivity, specificity and likelihood ratios.

As an example, considering PICC line as a risk factor of *Pseudomonas* bacteremia:

- A true positive was a patient with PICC line and *Pseudomonas* bacteremia

- A true negative was a patient with no PICC line and a bacteremia of an organism other than *Pseudomonas*

- A false positive was a patient with PICC line but a bacteremia with an organism other than *Pseudomonas*

- A false negative was a patient with no PICC line but *Pseudomonas* bacteremia.

We compared the negative likelihood ratio (NLR) of the predictors to the clinician's decision. A low NLR would be very useful for guiding empiric coverage, because it could be used to rule out MRSA or *Pseudomonas* bacteremia, such that the low risk patients could be safely spared broad-spectrum antibiotics with MRSA or *Pseudomonas* coverage.

All reported confidence intervals (CI) were two sided 95% interval and all tests were two sided with a $P<0.05$ significance level. All analyses were done with statistical software R 3.6.3 (Vienna, Austria).

## Results

### Description of bacteremia events

During the month of October in 2019, 201 bacteremia events occurred in 194 patients. Of the 201 bacteremia cases included in the study, the majority of cases occurred in elderly patients with multiple comorbidities (Table 2). Approximately half of the infections (47.3%) were acquired in the community.

For the 35 cases where the source was unknown at initiation of antibiotic therapy, the most commonly found infectious source later in the work-up were line associated bacteremia in 4 (11.4%) cases, intra-abdominal infection in 3 (8.6%) cases and pulmonary infection in 3 (8.6%) cases. Yet, at the initial work-up, of these 35 cases, 2 (5.7%) patients had an intravascular catheter from which cultures were not drawn, 25 (71.4%) had no abdominal imaging, and 5 (14.3%) had no chest imaging.

The most common organisms isolated from blood cultures included *E. coli* in 58 (28.9%) cases, *S. aureus* in 51 (25.4%) cases, *Enterococcus* in 22 (10.9%) cases and *P. aeruginosa* in 12 (6.0%) cases (Table 3). Of the 58 *E. coli* bacteremia cases, 13 (22.4%) had ESBL. Of the 51 *S. aureus* bacteremia cases, 23 (45.0%) were MRSA. The susceptibility profile for the 12 *P. aeruginosa* isolates is described in S1 Table.

### Empiric antibiotic choice

Of the 201 bacteremia cases, 109 (54.2%) cases received two or more empiric antibiotics. The most common antibiotics used were Vancomycin, Ceftriaxone and Piperacillin-Tazobactam (Table 3).

Vancomycin was used in 69 (34.3%) cases, but it was unnecessary in 48 (69.6%) of these cases. Empiric antibiotic therapy had *Pseudomonas* coverage in 107 (53.2%) cases, but it was unnecessary in 96 (89.7%) of these cases. Carbapenems was used in 31 (15.4%) cases, but it was unnecessary in 21 (67.7%) of these cases (S2 Table).

There were 100 (49.8%) cases where the hospital guidelines could be used to guide empiric antibiotic therapy based on presumed source at initiation of antibiotics. Of these 100 cases, 56 (56.0%) cases received empiric antibiotic therapy that was concordant with guideline recommendations. Of the 79 cases where the presumed source was indeed the infectious source and hospital guidelines were applicable, adherence to guidelines would result in adequate empiric antibiotics in 61 (77.2%) cases.

Table 2. Baseline characteristics and outcomes of patients with bacteremia.

| | Bacteremia events (N = 201) | A: Adequate antibiotic coverage (N = 145) | B: Inadequate antibiotic coverage (N = 56) | A vs. B P-value |
|---|---|---|---|---|
| Age median (IQR) | 67.0 (56.0, 75.0) | 65.0 (56.0, 74.0) | 70.5 (55.5, 76.0) | 0.174 |
| Female | 102 (50.8%) | 75 (51.7%) | 27 (48.2%) | 0.753 |
| Hospital site | | | | 0.233 |
| A | 56 (27.9%) | 38 (26.2%) | 18 (32.1%) | |
| B | 84 (41.8%) | 58 (40.0%) | 26 (46.4%) | |
| C | 61 (30.4%) | 49 (33.8%) | 12 (21.4%) | |
| Team that prescribed the empiric antibiotics | | | | 0.435 |
| Medicine | 109 (54.2%) | 81 (55.9%) | 28 (50.0%) | |
| ICU | 40 (19.9%) | 26 (17.9%) | 14 (25.0%) | |
| Surgery | 30 (14.9%) | 23 (15.9%) | 7 (12.5%) | |
| Hematology oncology | 18 (9.0%) | 11 (7.6%) | 7 (12.5%) | |
| ER | 4 (2.0%) | 4 (2.8%) | 0 (0%) | |
| Charlson comorbidity index | | | | 0.276 |
| 0 | 42 (20.9%) | 34 (23.5%) | 8 (14.3%) | |
| 1 | 34 (16.9%) | 23 (15.9%) | 11 (19.6%) | |
| 2 | 34 (16.9%) | 21 (14.5%) | 13 (23.2%) | |
| > = 3 | 91 (45.3%) | 67 (46.2%) | 24 (42.9%) | |
| Immunocompromised state | | | | |
| Chemotherapy | 27 (13.4%) | 22 (15.2%) | 5 (8.9%) | 0.355 |
| Steroid therapy | 22 (11.0%) | 13 (9.0%) | 9 (16.1%) | 0.205 |
| Neutropenia | 18 (9.0%) | 13 (9.0%) | 5 (8.9%) | >0.999 |
| Solid organ transplant | 7 (3.5%) | 3 (2.1%) | 4 (7.1%) | 0.096 |
| Bone marrow transplant | 10 (5.0%) | 7 (4.8%) | 3 (5.4%) | >0.999 |
| Risk factors for infection | | | | |
| Hemodialysis line | 13 (6.5%) | 9 (6.2%) | 4 (7.1%) | 0.758 |
| PICC line | 39 (19.4%) | 21 (14.5%) | 18 (32.1%) | 0.009 |
| Central line | 12 (6.0%) | 8 (5.5%) | 4 (7.1%) | 0.741 |
| Orthopedic prosthesis | 20 (10.0%) | 14 (9.7%) | 6 (10.7%) | 0.797 |
| Intracardiac device | 13 (6.5%) | 8 (5.5%) | 5 (8.9%) | 0.357 |
| Ventricular drain | 4 (2.0%) | 2 (1.4%) | 2 (3.6%) | 0.310 |
| Intra-abdominal drain | 10 (5.0%) | 7 (4.8%) | 3 (5.4%) | >0.999 |
| Urinary tract stent or drain | 7 (3.5%) | 5 (3.5%) | 2 (3.6%) | >0.999 |
| Injection drug use | 15 (7.5%) | 9 (6.2%) | 6 (10.7%) | 0.368 |
| Risk factors for antibiotic resistance | | | | |
| Last MRSA screening swab was positive | 20 (10.0%) | 11 (7.6%) | 9 (16.1%) | 0.111 |
| Last VRE screening swab was positive | 10 (5.0%) | 4 (2.8%) | 6 (10.7%) | 0.029 |
| Last ESBL screening swab was positive | 18 (9.0%) | 9 (6.2%) | 9 (16.1%) | 0.050 |
| Last CRE screening swab was positive | 2 (1.0%) | 1 (0.7%) | 1 (1.8%) | 0.481 |
| MRSA isolated from prior culture | 16 (8.0%) | 10 (6.9%) | 6 (10.7%) | 0.390 |
| VRE isolated from prior culture | 3 (1.5%) | 2 (1.4%) | 1 (1.8%) | >0.999 |
| ESBL isolated from prior culture | 9 (4.5%) | 5 (3.5%) | 4 (7.1%) | 0.268 |
| *Pseudomonas* isolated from prior culture | 19 (9.5%) | 10 (6.9%) | 9 (16.1%) | 0.060 |
| SPICE organism isolated form prior culture | 5 (2.5%) | 2 (1.4%) | 3 (5.4%) | 0.133 |
| Prior antibiotics within 90 days | 91 (45.3%) | 57 (39.3%) | 34 (60.7%) | 0.007 |
| Prior hospitalization within 30 days | 37 (18.4%) | 23 (15.9%) | 14 (25.0%) | 0.156 |

(*Continued*)

**Table 2.** (Continued)

| | Bacteremia events (N = 201) | A: Adequate antibiotic coverage (N = 145) | B: Inadequate antibiotic coverage (N = 56) | A vs. B P-value |
|---|---|---|---|---|
| Prior surgery within 90 days | 39 (19.4%) | 24 (16.6%) | 15 (26.8%) | 0.113 |
| Setting | | | | 0.043 |
| Community associated | 95 (47.3%) | 72 (49.7%) | 23 (41.1%) | |
| Healthcare associated | 39 (19.4%) | 32 (22.1%) | 7 (12.5%) | |
| Hospital associated | 54 (26.9%) | 35 (24.1%) | 19 (33.9%) | |
| ICU associated | 13 (6.5%) | 6 (4.1%) | 7 (12.5%) | |
| Source of infection | | | | 0.076 |
| Urinary tract infection | 46 (22.9%) | 38 (26.2%) | 8 (14.3%) | |
| Line associated infection | 37 (18.4%) | 23 (15.9%) | 14 (25.0%) | |
| Intra-abdominal infection | 30 (14.9%) | 21 (14.5%) | 9 (16.1%) | |
| Pulmonary infection | 19 (9.5%) | 16 (11.0%) | 3 (5.4%) | |
| Skin and soft tissue infection | 11 (5.5%) | 8 (5.5%) | 3 (5.4%) | |
| Other sources | 36 (17.9%)[a] | 28 (19.3%) | 8 (14.3%) | |
| Bacteremia without clear source | 22 (11.0%) | 11 (7.6%) | 11 (19.6%) | |
| Unknown source at initiation of antibiotic therapy | 35 (17.4%) | 19 (13.1%) | 16 (28.6%) | 0.013 |
| Pitt bacteremia score | | | | 0.759 |
| 0 | 125 (62.2%) | 91 (62.8%) | 34 (60.7%) | |
| 1 | 30 (14.9%) | 23 (15.9%) | 7 (12.5%) | |
| 2 | 15 (7.5%) | 11 (7.6%) | 4 (7.1%) | |
| > = 3 | 31 (15.4%) | 20 (13.8%) | 11 (19.6%) | |
| Hypotension | 46 (22.9%) | 33 (22.8%) | 13 (23.2%) | >0.999 |
| Peak lactate median (IQR) | 2.40 (1.50, 4.55) | 2.50 (1.60, 5.20) | 1.70 (1.25, 3.25) | 0.015 |
| Transfer to ICU | 53 (26.4%) | 36 (24.8%) | 17 (30.4%) | 0.476 |
| Persistent bacteremia | 39 (19.4%) | 27 (18.6%) | 12 (21.4%) | 0.692 |
| Recurrent bacteremia within 30 days | 6 (3.0%) | 3 (2.1%) | 3 (5.4%) | 0.351 |
| Death in 30 days | 38 (18.9%) | 25 (17.2%) | 13 (23.2%) | 0.317 |

CRE = carbapenem resistant enterobacteriaceae; ER = emergency room; ESBL = extended spectrum beta-lactamase; ICU = intensive care unit; IQR = interquartile range; MRSA = methicillin resistant *S. aureus*; PICC = peripherally inserted central catheter; SPICE = organisms that may have AmpC beta-lactamase including *Serratia*, *Providencia*, indole positive *Proteus*, *Citrobacter* and *Enterobacter* species; VRE = vancomycin resistant *Enterococcus*

[a]Other sources included bone or joint in 5 (2.5%) cases, CNS in 3 (1.5%) cases, decubitus ulcer in 4 (2.0%) cases, diabetic foot infection in 6 (3.0%) cases, endocarditis in 5 (2.5%) cases, endovascular source in 3 (1.5%) cases, ear/nose/throat in 4 (2.0%) cases, gynecological in 1 (0.5%) cases, ischemic foot in 1 (0.5%) cases, orthopedic prosthesis in 2 (1.0%) cases, and surgical wound in 2 (1.0%) cases

Of all 201 bacteremia cases, empiric coverage was adequate in 145 (72.1%) cases (Table 3). Of the 13 ESBL *E. coli* cases, 6 (46.2%) cases were treated with Piperacillin-Tazobactam. For all of these 6 cases, the isolates were tested to be susceptible to Piperacillin-Tazobactam with minimal inhibitory concentration (MIC) ≤8mg/L. Of the 12 SPICE organism cases, 2 (16.7%) cases were treated with Ceftriaxone. For these 2 cases, the isolates were tested to be susceptible to Ceftriaxone with MIC ≤0.25mg/L.

Potential predictors for inadequate empiric antibiotic therapy are listed in Table 4. In the final multivariable model, risk factors for inadequate empiric antibiotic coverage included unknown source at initiation of antibiotic therapy (adjusted odds ratio (aOR) of 2.76 95% CI 1.27–6.01, P = 0.010) and prior antibiotics within 90 days (aOR of 2.46 95% CI 1.30–4.74, P = 0.006).

**Table 3. Blood culture results and empiric antibiotic choice.**

| | Bacteremia events (N = 201) | A: Adequate antibiotic coverage (N = 145) | B: Inadequate antibiotic coverage (N = 56) | A vs. B P-value |
|---|---|---|---|---|
| Culture time to positivity in days median (IQR) | 0.8 (0.7, 1.1) | 0.8 (0.7, 1.0) | 0.9 (0.7, 1.5) | 0.002 |
| Culture time to susceptibility in days median (IQR) | 2.2 (1.8, 2.9) | 2.1 (1.7, 2.7) | 2.4 (1.9, 3.6) | 0.015 |
| Polymicrobial bacteremia | 51 (25.4%) | 30 (20.7%) | 21 (37.5%) | 0.019 |
| Common organisms isolated from blood culture | | | | |
| *E. coli* | 58 (28.9%) | 47 (32.4%) | 11 (19.6%) | 0.084 |
| *E. coli* that had ESBL | 13 (6.5%) | 6 (4.1%) | 7 (12.5%) | 0.050 |
| *S. aureus* | 51 (25.4%) | 40 (27.6%) | 11 (19.6%) | 0.282 |
| *S. aureus* that was methicillin resistant (MRSA) | 23 (11.4%) | 16 (11.0%) | 7 (12.5%) | 0.806 |
| *P. aeruginosa* | 12 (6.0%) | 6 (4.1%) | 6 (10.7%) | 0.098 |
| SPICE organisms | 12 (6.0%) | 1 (0.7%) | 11 (19.6%) | <0.001 |
| *E. faecalis* | 14 (7.0%) | 6 (4.1%) | 8 (14.3%) | 0.025 |
| *E. faecium* | 8 (4.0%) | 3 (2.1%) | 5 (8.9%) | 0.040 |
| *E faecium* that was vancomycin resistant (VRE) | 1 (0.5%) | 0 (0%) | 1 (1.8%) | 0.279 |
| Common empiric antibiotics used | | | | |
| Vancomycin | 69 (34.3%) | 63 (43.5%) | 6 (10.7%) | <0.001 |
| Ceftriaxone | 78 (38.8%) | 62 (42.8%) | 16 (28.6%) | 0.076 |
| Piperacillin-Tazobactam | 75 (37.3%) | 60 (41.4%) | 15 (26.8%) | 0.073 |
| Meropenem | 25 (12.4%) | 18 (12.4%) | 7 (12.5%) | >0.999 |
| Ertapenem | 6 (3.0%) | 2 (1.4%) | 4 (7.1%) | 0.052 |
| Empiric *Pseudomonas* coverage | 107 (53.2%) | 81 (55.9%) | 26 (46.4%) | 0.270 |
| Unnecessary *Pseudomonas* coverage | 96 (47.8%) | 75 (51.7%) | 21 (37.5%) | 0.084 |
| Unnecessary use of carbapenem | 21 (10.5%) | 14 (9.7%) | 7 (12.5%) | 0.609 |
| Unnecessary use of Vancomycin | 48 (23.9%) | 42 (29.0%) | 6 (10.7%) | 0.006 |

IQR = interquartile range; MRSA = methicillin resistant *S. aureus*; PICC = peripherally inserted central catheter; SPICE = organisms that may have AmpC beta-lactamase including *Serratia*, *Providencia*, indole positive *Proteus*, *Citrobacter* and *Enterobacter* species; VRE = vancomycin resistant *Enterococcus*

## Comparison of clinical pathway to clinician's decision on empiric MRSA and *Pseudomonas* coverage

Patient risk factors considered for a clinical pathway predicting MRSA included hemodialysis line, injection drug use, positive MRSA screening swab and MRSA isolated from prior culture (Table 5). However, the clinician's decision for empiric Vancomycin was better than any of the above risk factors based on having the lowest NLR of 0.43.

Patient risk factors considered for a clinical pathway predicting *Pseudomonas* bacteremia included neutropenia, PICC line, *Pseudomonas* isolated from prior culture, prior antibiotics within 90 days and hospital or healthcare associated infection (Table 6). A clinical pathway that considered hospital or healthcare associated infection to be at high risk for *Pseudomonas* was better at predicting *Pseudomonas* bacteremia than the clinician's decision on empiric *Pseudomonas* coverage based on a lower NLR of 0.17 versus 0.34. If only hospital or healthcare associated bacteremia cases were given antibiotics with *Pseudomonas* coverage, there would be 39 less patients on antibiotics with *Pseudomonas* coverage that did not have *Pseudomonas*

**Table 4. Univariate analysis of predictors of inadequate empiric antibiotic therapy.**

| Predictors | Odds ratio (95% CI) | P-value |
|---|---|---|
| Team that prescribed the empiric antibiotics | | |
| Medicine | Reference | |
| ICU | 1.73 (0.79–3.74) | 0.161 |
| Surgery | 0.72 (0.25–1.85) | 0.522 |
| Hematology oncology | 1.84 (0.62–5.15) | 0.250 |
| Setting | | |
| Community associated | Reference | |
| Healthcare associated | 0.68 (0.25–1.69) | 0.431 |
| Hospital associated | 1.57 (0.75–3.27) | 0.232 |
| ICU associated | 5.01 (1.52–18.04) | 0.009 |
| Unknown source at initiation of antibiotic therapy | 2.65 (1.24–5.65) | 0.011 |
| Prior antibiotics within 90 days | 2.39 (1.28–4.53) | 0.007 |
| Prior surgery within 90 days | 1.60 (0.75–3.33) | 0.215 |

CI = confidence interval

bacteremia. As well, there would be 1 more patient with *Pseudomonas* bacteremia on antibiotics with *Pseudomonas* coverage compared to what the clinician prescribed.

The decrease in probability of *Pseudomonas* bacteremia based on the clinician's decision or community associated infection is illustrated in S1 Fig. As an example, if the pretest probability of *Pseudomonas* bacteremia were 10%, the clinician's impression of being at low risk for *Pseudomonas* would decrease the post-test probability of *Pseudomonas* bacteremia to approximately 4% (95% CI 1%-12%) whereas classifying community associated infection as low risk would decrease the post-test probability to approximately 2% (95% CI 0.3%-11%).

We have explored pathways using a combination of the risk factors for *Pseudomonas* bacteremia, which did not significantly improve the diagnostic accuracy and had a low number of negative cases such that it was not clinically useful (S3 Table).

## Discussion

In this retrospective cohort study of 201 bacteremia cases, clinician's empiric antibiotic coverage was adequate in 72% cases. Risk factors for inadequate empiric coverage included

**Table 5. Diagnostic accuracy of patient risk factors and clinician's decision on empiric Vancomycin in predicting MRSA bacteremia.**

| | Sensitivity (95% CI) | Specificity (95% CI) | PLR (95% CI) | NLR (95% CI) |
|---|---|---|---|---|
| Hemodialysis line | 0.22 (0.10–0.42) | 0.96 (0.91–0.98) | 4.84 (1.73–13.54) | 0.82 (0.66–1.02) |
| Injection drug use | 0.22 (0.10–0.42) | 0.94 (0.90–0.97) | 3.87 (1.45–10.33) | 0.83 (0.67–1.03) |
| Last MRSA screening swab was positive | 0.52 (0.33–0.71) | 0.65 (0.58–0.72) | 1.50 (0.97–2.33) | 0.73 (0.47–1.14) |
| MRSA isolated from prior culture | 0.30 (0.16–0.51) | 0.95 (0.91–0.97) | 6.02 (2.48–14.62) | 0.73 (0.56–0.96) |
| Clinician's decision to cover MRSA empirically | 0.70 (0.49–0.84) | 0.70 (0.63–0.77) | 2.34 (1.64–3.32) | 0.43 (0.23–0.81) |

CI = confidence interval; MRSA = methicillin resistant *S. aureus*; NLR = negative likelihood ratio; PLR = positive likelihood ratio

**Table 6. Diagnostic accuracy of patient risk factors and clinician's decision on empiric *Pseudomonas* coverage in predicting *Pseudomonas* bacteremia.**

|  | Sensitivity | Specificity | PLR | NLR |
|---|---|---|---|---|
| Neutropenia | 0.25 (0.09–0.53) | 0.92 (0.87–0.95) | 3.15 (1.06–9.40) | 0.82 (0.59–1.13) |
| PICC line | 0.50 (0.25–0.75) | 0.83 (0.77–0.87) | 2.86 (1.50–5.46) | 0.61 (0.34–1.07) |
| *Pseudomonas* isolated from prior culture | 0.50 (0.25–0.75) | 0.93 (0.89–0.96) | 7.27 (3.36–15.72) | 0.54 (0.31–0.95) |
| Prior antibiotics within 90 days | 0.83 (0.55–0.95) | 0.57 (0.50–0.64) | 1.94 (1.44–2.63) | 0.29 (0.08–1.04) |
| Hospital or healthcare associated | 0.92 (0.65–1.00) | 0.50 (0.43–0.57) | 1.82 (1.46–2.28) | 0.17 (0.03–1.10) |
| Clinician's decision to cover *Pseudomonas* empirically | 0.83 (0.55–0.95) | 0.49 (0.42–0.56) | 1.62 (1.22–2.17) | 0.34 (0.10–1.22) |

CI = confidence interval; MRSA = methicillin resistant *S. aureus*; NLR = negative likelihood ratio; PICC = peripherally inserted central catheter; PLR = positive likelihood ratio

unknown source at initiation of antibiotic therapy and prior antibiotics within 90 days. Clinician's decision for empiric MRSA coverage was better than any patient MRSA risk factor. However, a clinical pathway that considered hospital or healthcare associated infection to be at high risk for *Pseudomonas* was more accurate than the clinician's decision in predicting when to empirically cover for *Pseudomonas*.

The proportion of inadequate empiric coverage at 28% in our study was close to the pooled estimate of 32% in a systematic review of studies on inadequate empiric antibiotic therapy [7]. The heterogeneity of this estimate across different studies may be attributed to different definitions for empiric and adequate antibiotic therapy across studies [7, 23].

In terms of risk factors for inadequate empiric coverage found in our study, prior antibiotic therapy is an important predictor for antibiotic resistance in the current infection [24], likely because it selects for more resistant organisms in subsequent infections. This increases the probability of the isolate being resistant to the empiric antibiotic therapy. When a source is known at initiation of antibiotics, clinicians have a better estimate of probable pathogens to cover based on the syndrome and clinical guidelines, so they are more likely to choose adequate empiric antibiotic coverage. Conversely, in cases where the source is unknown at initiation of empiric antibiotics, it is likely to be associated with inadequate empiric coverage as shown previously [25].

In our study, a simple clinical pathway based on a patient risk factor may be more accurate for predicting when to empirically cover for *Pseudomonas* than the clinician's decisions. Similarly, patient risk factors such as prior culture and susceptibility have been incorporated in decision support models and may help improve the accuracy of empiric antibiotic coverage [26].

Our study findings offer an example of a way to use local audit data of empiric antibiotic therapy to identify areas of improvement for clinician's empiric antibiotic choice. Based on the data, we could provide the following specific feedback to clinicians at the study hospitals that may improve empiric antibiotic prescribing in the future. First, clinicians should consider resistant organisms for patients who received prior antibiotic therapy and do a thorough diagnostic work-up to determine the infectious source, as these were the common pitfalls for inadequate empiric coverage in the audit. For example, an important number of patients with unknown source at the initiation of antibiotic therapy in the audit did not have a blood culture drawn from the intravascular catheter, chest imaging, or abdominal imaging despite these being the most common infectious sources in this group. Second, although clinicians follow

hospital guidelines for empiric antibiotic therapy in the majority of cases, these guidelines were only applicable in approximately half of bacteremia cases and a significant proportion of patients had an unknown source at initiation of empiric antibiotics. Therefore, a guideline on empiric antibiotic therapy for patients with sepsis and suspected bacteremia without a clear source may be helpful to guide clinicians. On this guideline, clinicians should consider whether the infection is hospital or healthcare associated before deciding on empiric *Pseudomonas* coverage. Patients with community associated infection likely do not need *Pseudomonas* coverage. This would be applicable to the hospitals where the study was done with the caveat that it may not apply to all clinical scenarios. There are many practical ways to bring these audit-based feedbacks to the real world hospital setting. For example, a simple clinical pathway for empiric *Pseudomonas* coverage could be built in the electronic health record that could be linked to orders of blood culture for suspected bacteremia. A pharmacist could review inpatients with blood cultures in progress that are on empiric Vancomycin, antibiotics with *Pseudomonas* coverage or carbapenems on a daily basis. During this review, the pharmacist can provide recommendations on whether these broad-spectrum empiric antibiotics are necessary based on hospital guidelines and patient risk factors.

This study has several strengths. First, we used a strict definition for adequate empiric coverage considering the exact time of antibiotic initiation, dose, route and bioavailability. Second, we had detailed patient information regarding risk factors for antibiotic resistant organisms such as prior culture and screening swabs. Lastly, we considered the patient's infectious source and isolates from cultures other than blood cultures before considering an antibiotic coverage unnecessary.

The study also has several limitations. First, antibiotic resistance and empiric antibiotic prescribing patterns vary greatly across different hospitals. Therefore, the predictors of inadequate empiric coverage or antibiotic resistant organisms may not apply to settings outside the study. Our study is not meant to derive predictors to be applied to other hospitals. Rather, our study was meant to demonstrate the use of local data to provide guidance on how to optimize clinician's empiric antibiotic selection. Instead of applying the predictors from our study, we would encourage local hospitals to undergo similar auditing and analysis process to identify areas for improvement.

Second, the study sample was relatively small compared to other multicenter studies on empiric antibiotic therapy [7, 27]. Local hospitals cannot undertake such a large study for quality improvement projects. Our study illustrates how a study with a smaller sample size was more feasible and still provided meaningful results that could be used to improve empiric antibiotic use. It should be noted that smaller facilities might not have the resources or analytical expertise to perform such analyses. In low resource settings, we believe that crude data alone from an audit could still be helpful. Future quasi-experimental studies should determine whether the data derived feedback leads to an increase in adequate empiric antibiotic coverage for bacteremia.

Third, we tried to simulate the empiric antibiotic decision on which pathogens to cover using diagnostic accuracy parameters including positive likelihood ratios (PLR) and NLR. Based on the reported PLR and NLR of the reported clinical pathways, it seemed that ruling in a pathogen using a clinical pathway was not helpful due to the low PLR values. It should be noted that use of diagnostic properties is an over-simplification of a complex decision by the clinician that considered many factors simultaneously.

## Conclusion

In conclusion, we performed an audit of empiric antibiotic therapy for bacteremia, which showed risk factors for inadequate empiric coverage as well as a clinical pathway based on a

patient risk factor that better predicts when to cover for *Pseudomonas* bacteremia than the clinician's decision. Therefore, audit of antibiotic therapy in bacteremia is feasible and may provide useful feedback on how to locally improve empiric antibiotic therapy.

## Supporting information

**S1 Fig.**
(EPS)

**S1 Table.**
(DOCX)

**S2 Table.**
(DOCX)

**S3 Table.**
(DOCX)

**S1 Data. Deidentified data set.**
(CSV)

## Author Contributions

**Conceptualization:** Anthony D. Bai, Neal Irfan, Cheryl Main, Philippe El-Helou, Dominik Mertz.

**Data curation:** Anthony D. Bai, Cheryl Main.

**Formal analysis:** Anthony D. Bai.

**Investigation:** Anthony D. Bai.

**Methodology:** Anthony D. Bai, Dominik Mertz.

**Project administration:** Anthony D. Bai.

**Supervision:** Dominik Mertz.

**Writing – original draft:** Anthony D. Bai, Neal Irfan, Cheryl Main, Philippe El-Helou, Dominik Mertz.

**Writing – review & editing:** Anthony D. Bai, Neal Irfan, Cheryl Main, Philippe El-Helou, Dominik Mertz.

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
