## [Decision Letter · Decision Letter 0]

21 Jan 2021

PONE-D-20-40076

Can a local audit of antibiotic therapy in bacteremia provide useful feedback on how to improve clinicians’ empiric antibiotic choice? A retrospective cohort study

PLOS ONE

Dear Dr. Bai,

Thank you for submitting your manuscript to PLOS ONE. After careful consideration, we feel that it has merit but does not fully meet PLOS ONE’s publication criteria as it currently stands. Therefore, we invite you to submit a revised version of the manuscript that addresses the points raised during the review process.

The reviewers have carefully evaluated the manuscript and commented in details. Please follow their comments and address them one by one. Make sure you terminology regarding therapy is consistent throughout the manuscript. Please provide details regarding coverage of empiric antibiotics that is insufficient and coverage that is excessively broad (superfluous). Make sure the title represents what is presented in the study. Make sure the limited external validity of the study is mentioned, as the reviewers recommended. 

We look forward to receiving your revised manuscript.

Kind regards,

Dafna Yahav

Academic Editor

PLOS ONE

2. Thank you for stating in your ethics statement "Consent was not obtained as the study was a retrospective chart review and the data were analyzed anonymously." In your Methods section and the ethics statement in the online submission form, please clarify whether all data were also fully anonymized before you accessed them, and/or whether the IRB or ethics committee waived the requirement for informed consent.

3. Thank you for providing the date(s) when patient medical information was initially recorded. Please also include the date(s) on which your research team accessed the databases/records to obtain the retrospective data used in your study.

Reviewers' comments:

Reviewer's Responses to Questions

**Comments to the Author**

1. Is the manuscript technically sound, and do the data support the conclusions?

Reviewer #1: Yes

Reviewer #2: Partly

2. Has the statistical analysis been performed appropriately and rigorously? 

Reviewer #1: Yes

Reviewer #2: Yes

3. Have the authors made all data underlying the findings in their manuscript fully available?

Reviewer #1: No

Reviewer #2: No

4. Is the manuscript presented in an intelligible fashion and written in standard English?

Reviewer #1: Yes

Reviewer #2: Yes

5. Review Comments to the Author

Reviewer #1: The authors performed a cohort study of patients receiving antibiotic therapy for positive blood cultures and evaluate adequacy of coverage and risk factors for inadequate coverage. This is an important endeavor for antimicrobial stewardship programs as empiric appropriateness may often be overlooked because it is more feasible to review antibiotics later during the stay and/or evaluate overall antibiotic utilization data. The use of terminology to describe appropriateness/adequacy in this manuscript was somewhat confusing and could be standardized further. Since inappropriate antibiotic therapy usually includes both inadequate (coverage of empiric antibiotics is insufficient) and unnecessary (coverage is excessively broad), both should be addressed and appropriate terminology should be used throughout.

The following are some considerations for improvement:

ABSTRACT

1. "Correct" terminology implies that excessively broad empiric antibiotics are appropriate. Consider using terminology "adequate" and "inadequate" or similar.

Consider using a framework similar to that of NAPS for clear/standardized definitions: https://www.safetyandquality.gov.au/sites/default/files/2020-02/report_-_2018_hospital_naps.pdf (page 49)

2. Consider providing a difference between negative likelihood ratios for the clinical pathway vs. clinician decision with a confidence interval around the point estimate. Although numerically different, the confidence interval for both likelihood ratios appears quite wide and hard to appreciate a difference between the estimates.

INTRODUCTION

3. As above, please reconsider the use of the terms "correct" and "incorrect" coverage.

METHODS

4. Were only inpatients included, or could non-admitted patients also count (e.g., blood culture in ED)?

5. Were Staphylococcus lugdunensis cases excluded from analysis, as they would be part of CNST category?

6. How were polymicrobial blood cultures handled?

7. Since patient data were collected retrospectively, and patient status or information may change during the hospital stay, please indicate whether these data were from the empiric period (e.g., infectious source - suspected source could shift over time, severity of infection - this could also change over time) or from any point during the hospital stay.

8. A brief statement and/or reference may be helpful to justify why 24 hours was selected as the empiric period (e.g., as opposed to a longer duration). Does the hospital use any form of rapid bacteriological testing?

9. Correct empiric coverage lines 131-157 use various terms "appropriate" and "unnecessary". Suggest using consistent terminology throughout the manuscript. It was not clear until now that unnecessarily broad empiric therapy would be considered. It appears that the authors do not count this as incorrect, but rather a separate category that is not necessarily mutually exclusive. It is somewhat confusing that an antibiotic could be both "correct" but "unnecessary". Since unnecessary therapy (e.g., vancomycin when not needed) is different than inadequate therapy (e.g., no vancomycin when it is needed), these should be more clearly distinguished throughout.

10. Line 164 "team that administered antibiotics" is unclear. Does this mean the medical or surgical service that prescribed the antibiotic?

RESULTS

11. Consider adding 2 columns to Table 2 and/or 3 to show characteristics of "inadequate" and "inadequate" treatment events, as this may be helpful to visually highlight crude differences in the population (in addition to the multivariable model shown later).

12. Table 4 title indicates "inappropriate" empiric therapy. It is not clear if it refers to adequacy of coverage or unnecessary coverage or both?

13. Many antimicrobial stewards will be interested in the total proportion of inappropriate antibiotics ordered (accounting for both incorrect or inadequate coverage and unnecessary coverage). As such, the current % "correct" antibiotic coverage likely underestimates inappropriateness. Has the total inappropriate % been calculated?

DISCUSSION

13. Although useful, many hospitals particularly smaller facilities, will not have the resources or analytical expertise to perform such analyses and identify opportunities for improvement. This should be acknowledged or comments provided to consider how low resource settings would perform such an audit (e.g., is using crude data alone enough?).

Reviewer #2: The authors describe the results of an audit of antibiotic prescription practice and present a new diagnostic "tool" to improve antibiotic use (mainly to reduce un-necessary broad-spectrum antibiotic use). The main findings are that 72% of the empiric antibiotic courses were incorrect, and they have pointed out the variables associated with incorrect empiric treatment, which included prior antibiotic use, and unknown source of infection. The diagnostic accuracy tables are presented but their impact on antibiotic use is not described in this manuscript, neither the outcome of correct versus incorrect empiric treatment.

Comments:

1. The title is confusing - the study only demonstrates the results of a local audit and does not really answer the question presented in the title. Consider modifying it to reflect the study objectives and findings.

2. The univariable analysis includes only a small number of variables and only odds ratio are presented. I would be interested in a table that compares patients who were and were not treated correctly. This table should include variables that may be associated with correct empiric treatment, including many of the variables presented in Table 2 and 3. I would also recommend to add the antibiotic treatment that were used in such a table, as well as the pathogens treated and their resistance patterns.

3. Using diagnostic accuracy parameters (positive and negative likelihood ratios, PLR, NLR, respectively) is a nice idea, but when treating life threatening bloodstream infection (BSI), I believe that the PLR should be higher in any given diagnostic test than the one presented here, this should be addressed in the discussion.

In this case the results are only relevant to rule out each of the BSI evaluared. Taking Pseudomonas BSI as an example, 50% (6/12) of patients with pseudomonas BSI did not receive empiric antipseudomonal coverage. According to the results, using NLR 0.17 of healthcare or hospital infection means that if patients did not have hospital or healthcare associated infection then 39 patients would not receive anti-pseudomonal coverage and additional 1 patient would receive anti-pseudomonal coverage if the patients had hospital acquired infection.

In the discussion the authors claim that not having hospital acquired infection means that the patient will likely not need anti-pseudomonal coverage, but it really cannot be a stand-alone clinical decision tool. First, it is unkown from this study results the impact of using this diagnostic tool on clinical outcomes. Second, as the authors well noted, the results of this study cannot be used in other hospitals and external validation should be performed.

Finally, I think that the authors should address the limitations of using diagnostic accuracy tables for clinical decisions and provide more data about how combined risk factors modified NLR and PLR, even if only as a supplemental table.

Clinician's decision – are there any criteria that clinicians used? were there more than one clinician that evaluated these cases? it is very reasonable that clinicians used the exact same risk factors to reach decision from the same table, and these are very heterogeneous decisions. That's why I don't think it should be used to evaluate diagnostic accuracy.

Minor comments:

How were prior cultures defined? What was the time period before the BSI?

The terms "correct" or "appropriate" should be used uniformly and not interchangeably.

Can the authors describe what were the antibiotics that were provided inappropriately? for example, how many ESBL pathogens were covered by piperacillin-tazobactam and thus defined inappropriately treated (how many were susceptible to piperacillin-tazobactam)? how many SPICE pathogens were empirically covered by third generation cephalosporins and defined as inappropriate empiric therapy (how many were susceptible to third generation cephalosporins)?

Table 4 – remove ER

Page 14, line 216 – the fact that blood cultures were not drawn from the intravascular catheter does not preclude it from being the source of infection. I would not categorized it as "unknown source".

Table 2: please describe "other sources" in the footnotes.

Can the authors describe what were the cases where carbapenem was used un-necessarily?

What was the antibiogram of Pseudomonas bacteremia? how many of these BSI were susceptible to each antibiotics?

In the discussion would address the fact that guidelines were applicable only for ~40% of the cases.

I believe it would be helpful to the readers if the authors could demonstrate how the NLR can be applied by showing pre-test and post-test probabilities for each risk factor and associated PLR and NLR.

6. PLOS authors have the option to publish the peer review history of their article (what does this mean?). If published, this will include your full peer review and any attached files.

Reviewer #1: No

Reviewer #2: No

---

## [Author Response · Author response to Decision Letter 0]

5 Feb 2021

Journal Requirements

Comment #1: Please ensure that your manuscript meets PLOS ONE's style requirements, including those for file naming.

Response #1: We have reviewed the PLOS ONE style requirements and made the necessary edits to meet these requirements. 

Comment #2: Thank you for stating in your ethics statement "Consent was not obtained as the study was a retrospective chart review and the data were analyzed anonymously." In your Methods section and the ethics statement in the online submission form, please clarify whether all data were also fully anonymized before you accessed them, and/or whether the IRB or ethics committee waived the requirement for informed consent.

Response #2: The data was anonymized after we accessed the data. The research ethics committee waived the requirement for informed consent. We have added this description to the manuscript (please see Methods, sub-heading Study design, 1st paragraph, 2nd sentence and Methods, sub-heading Data collection, 1st paragraph, 4th sentence). 

Comment #3: Thank you for providing the date(s) when patient medical information was initially recorded. Please also include the date(s) on which your research team accessed the databases/records to obtain the retrospective data used in your study.

Response #3: The patient medical information was all recorded on the same day of the patient encounter. Our team accessed the patient electronic medical record from October 14, 2020 to Dec 7, 2020. This is now described in the manuscript (please see Methods, sub-heading Data collection, 1st paragraph, 1st and 2nd sentences).

Comment #4: We note that you have indicated that data from this study are available upon request. PLOS only allows data to be available upon request if there are legal or ethical restrictions on sharing data publicly. 

Response #4: We have attached a de-identified dataset as a supplemental file. As per the attached BMJ reference, we deleted confidential information such as age, sex and hospital site from this dataset. 

 

Reviewer 1 Comments

Comment #1: The use of terminology to describe appropriateness/adequacy in this manuscript was somewhat confusing and could be standardized further. Since inappropriate antibiotic therapy usually includes both inadequate (coverage of empiric antibiotics is insufficient) and unnecessary (coverage is excessively broad), both should be addressed and appropriate terminology should be used throughout. . "Correct" terminology implies that excessively broad empiric antibiotics are appropriate. Consider using terminology "adequate" and "inadequate" or similar.

Response #1: We agree with the reviewer that adequate and inadequate are better descriptions of the antibiotic coverage of the organisms isolated in the blood culture. We have changed the term from “correct” and “incorrect” to “adequate” and “inadequate” coverage throughout the manuscript. We also ensured that this terminology remained consistent throughout the manuscript. 

Comment #2: Consider providing a difference between negative likelihood ratios for the clinical pathway vs. clinician decision with a confidence interval around the point estimate. Although numerically different, the confidence interval for both likelihood ratios appears quite wide and hard to appreciate a difference between the estimates.

Response #2: We agree with the reviewer that it may be difficult to appreciate the difference in the likelihood ratios. However, we do not think a numerical difference of the likelihood ratios is meaningful, because likelihood ratio describe relative to the odds of disease and are not based on a numerical linear scale. In order to help the reader interpret this better, we added a figure (please see S2 Supplemental Figure 1) to illustrate the difference with confidence interval in post-test probability between the clinician’s decision and clinical pathway. 

Comment #3: As above, please reconsider the use of the terms "correct" and "incorrect" coverage.

Response #3: We have changed the term from “correct” and “incorrect” to “adequate” and “inadequate” coverage throughout the manuscript.

Comment #4: Were only inpatients included, or could non-admitted patients also count (e.g., blood culture in ED)?

Response #4: Non-admitted patients in the ER and inpatients were included in the study. We have clarified this in the manuscript (please see Methods, sub-heading Patient selection, 1st paragraph, 2nd sentence). 

Comment #5: Were Staphylococcus lugdunensis cases excluded from analysis, as they would be part of CNST category?

Response #5: No, Staphylococcus lugdunensis was the exception that would be included in the analysis. We clarified this in the exclusion criteria (please see Methods, sub-heading Patient selection, 2nd paragraph, 2nd sentence). 

Comment #6: How were polymicrobial blood cultures handled?

Response #6: Polymicrobial blood cultures could be included as long as patients do not meet the exclusion criteria. In these cases, the empiric antibiotic therapy was considered adequate if the antibiotic therapy covered all of the organisms grown in the blood culture. We have added this clarification in the manuscript (please see Methods, sub-heading Adequate empiric coverage, 2nd paragraph, 2nd sentence). 

Comment #7: Since patient data were collected retrospectively, and patient status or information may change during the hospital stay, please indicate whether these data were from the empiric period (e.g., infectious source - suspected source could shift over time, severity of infection - this could also change over time) or from any point during the hospital stay.

Response #7: The severity of infection was within first 24 hours of blood culture collection. The suspected infectious source was based on the documentation when empiric antibiotics were started and the infectious source was determined at the end of hospital stay. We have clarified these time points in the manuscript (please see Methods, sub-heading Data collection, 1st paragraph, 3rd sentence). 

Comment #8: A brief statement and/or reference may be helpful to justify why 24 hours was selected as the empiric period (e.g., as opposed to a longer duration). Does the hospital use any form of rapid bacteriological testing?

Response #8: As per the reviewer’s suggestion, we have added justification based on references for 24 hours as the time point to define empiric period (please see Methods, sub-heading Adequate empiric coverage, 1st paragraph, 2nd and 3rd sentence). The microbiology does not use any form of rapid bacteriological testing besides MALDI-TOF for identification of species. This is now added to the manuscript (please see Methods, sub-heading Adequate empiric coverage, 3rd paragraph, 1st sentence). 

Comment #9: Correct empiric coverage lines 131-157 use various terms "appropriate" and "unnecessary". Suggest using consistent terminology throughout the manuscript. It was not clear until now that unnecessarily broad empiric therapy would be considered. It appears that the authors do not count this as incorrect, but rather a separate category that is not necessarily mutually exclusive. It is somewhat confusing that an antibiotic could be both "correct" but "unnecessary". Since unnecessary therapy (e.g., vancomycin when not needed) is different than inadequate therapy (e.g., no vancomycin when it is needed), these should be more clearly distinguished throughout.

Response #9: With the change of terminology to “adequate” and “inadequate” empiric coverage in the revised manuscript, this should be less confusing. We added explanation to clarify that unnecessary broad antibiotic overage was considered separate from adequate coverage. We also added examples to illustrate cases where an adequate antibiotic therapy could still be unnecessarily broad (please see Methods, sub-heading Adequate empiric coverage, 6th paragraph, 1st, 5th to 8th sentences). 

Comment #10: Line 164 "team that administered antibiotics" is unclear. Does this mean the medical or surgical service that prescribed the antibiotic?

Response #10: Yes, we meant the service that prescribed the antibiotic. We have changed from “team that administered the empiric antibiotics” to “team that prescribed the empiric antibiotics” (please see Table 2). 

Comment #11: Consider adding 2 columns to Table 2 and/or 3 to show characteristics of "inadequate" and "inadequate" treatment events, as this may be helpful to visually highlight crude differences in the population (in addition to the multivariable model shown later).

Response #11: As the reviewer suggested, we have added the description and comparison of adequate and inadequate empiric coverage groups on Table 2 and 3 (please see Table 2 and 3).

Comment #12: Table 4 title indicates "inappropriate" empiric therapy. It is not clear if it refers to adequacy of coverage or unnecessary coverage or both?

Response #12: This was our mistake. We meant adequacy of coverage. We have changed the title to “inadequate” empiric therapy (please see Table 4 title). 

Comment #13: Many antimicrobial stewards will be interested in the total proportion of inappropriate antibiotics ordered (accounting for both incorrect or inadequate coverage and unnecessary coverage). As such, the current % "correct" antibiotic coverage likely underestimates inappropriateness. Has the total inappropriate % been calculated?

Response #13: We agree that it would be interesting to know the proportion of inappropriate antibiotic therapy accounting for both adequacy of coverage as well as excessively broad coverage. However, we think that this appropriateness would be difficult to define and may vary on a case by case basis depending on the clinical scenario accounting for the other factors such as infectious source, antibiotic allergy etc. Therefore, we think this is outside the scope of this study and did not attempt to define appropriateness / inappropriateness. 

Comment #14: Although useful, many hospitals particularly smaller facilities, will not have the resources or analytical expertise to perform such analyses and identify opportunities for improvement. This should be acknowledged or comments provided to consider how low resource settings would perform such an audit (e.g., is using crude data alone enough?).

Response #14: The reviewer makes a good point that smaller facilities may not be able to perform such analyses. Within the manuscript, we added the acknowledgement that such undertaking may not be feasible in low resource setting and we believe that an audit with crude data may still be helpful (please see Discussion, 8th paragraph, 4th and 5th sentences). 

 

Reviewer 2 Comments

Comment #1: The title is confusing - the study only demonstrates the results of a local audit and does not really answer the question presented in the title. Consider modifying it to reflect the study objectives and findings.

Response #1: The reviewer makes a good point that the title was misleading. We have changed the title to “Local audit of empiric antibiotic therapy in bacteremia: A retrospective cohort study”. 

Comment #2: The univariable analysis includes only a small number of variables and only odds ratio are presented. I would be interested in a table that compares patients who were and were not treated correctly. This table should include variables that may be associated with correct empiric treatment, including many of the variables presented in Table 2 and 3. I would also recommend to add the antibiotic treatment that were used in such a table, as well as the pathogens treated and their resistance patterns.

Response #2: As the reviewer suggested, we have added the description and comparison of adequate and inadequate empiric coverage groups on Table 2 and 3 (please see Table 2 and 3). Table 3 describes the common antibiotics used and the common resistant pathogens such as ESBL and MRSA. We believe that this level of detail would be adequate. Any further detail would not be practical to be displayed in a table considering the number of pathogens and number of antibiotics being tested for susceptibility. The reader can access the dataset with individual patient information for this. 

Comment #3: Using diagnostic accuracy parameters (positive and negative likelihood ratios, PLR, NLR, respectively) is a nice idea, but when treating life threatening bloodstream infection (BSI), I believe that the PLR should be higher in any given diagnostic test than the one presented here, this should be addressed in the discussion.

In this case the results are only relevant to rule out each of the BSI evaluared. Taking Pseudomonas BSI as an example, 50% (6/12) of patients with pseudomonas BSI did not receive empiric antipseudomonal coverage. According to the results, using NLR 0.17 of healthcare or hospital infection means that if patients did not have hospital or healthcare associated infection then 39 patients would not receive anti-pseudomonal coverage and additional 1 patient would receive anti-pseudomonal coverage if the patients had hospital acquired infection.

Response #3: The reviewer brings up an important limitation. We added the poor PLR being not clinically useful as a limitation in the manuscript (please see Discussion, 9th paragraph, 1st and 2nd sentence). 

Comment #4: In the discussion the authors claim that not having hospital acquired infection means that the patient will likely not need anti-pseudomonal coverage, but it really cannot be a stand-alone clinical decision tool. First, it is unknown from this study results the impact of using this diagnostic tool on clinical outcomes. Second, as the authors well noted, the results of this study cannot be used in other hospitals and external validation should be performed.

Response #4: We agree with both of the reviewer’s points. We added emphasis that this finding would only be applicable to the hospitals where the study was done and that there are always exceptions to this rule (please see Discussion, 5th paragraph, 9th sentence). 

Comment #5: Finally, I think that the authors should address the limitations of using diagnostic accuracy tables for clinical decisions and provide more data about how combined risk factors modified NLR and PLR, even if only as a supplemental table.

Response #5: We added a description of the limitation of using diagnostic accuracy properties (please see Discussion, 9th paragraph). We also added a supplemental table to show no benefit of combining risk factors in terms of improving the PLR and NLR for Pseudomonas bacteremia (please see S1 Table 3). 

Comment #6: Clinician's decision – are there any criteria that clinicians used? were there more than one clinician that evaluated these cases? it is very reasonable that clinicians used the exact same risk factors to reach decision from the same table, and these are very heterogeneous decisions. That's why I don't think it should be used to evaluate diagnostic accuracy.

Response #7: The clinician who took care of the patient makes the decision on his or her own. They have access to resources including the site-specific antibiogram and the hospital guideline recommendations on empiric antibiotic therapy. This was explained in more detail in the manuscript (please see Methods, sub-heading Hospital policy, 1st paragraph, 1st and 2nd sentences). The reviewer makes a good point that clinicians may consider the exact same risk factors in the same table. However, the factors considered is likely not standardized across clinicians. For example, if the clinician truly considered all of the risk factors in the table, then the clinician’s decision should be as good, if not better than any individual risk factor listed on the table. However, that is not the case, because how the infection was acquired had a better diagnostic properties than the clinician’s decision, so at least some clinicians must not have accounted for or disregarded the importance of acquisition. If a simple pathway such as risk stratifying based on acquisition can standardize decision making that is shown to be better than the heterogeneous decision that clinicians make, then maybe some empiric antibiotic choices should not be left solely to the clinician’s judgment and decision that is very heterogeneous across clinicians. 

Comment #8: How were prior cultures defined? What was the time period before the BSI?

Response #8: We defined prior cultures as being within 12 months before the bacteremia episodes. We added this clarification in the manuscript (please see Methods, sub-heading Definition of variables, 3rd paragraph, 1st sentence). 

Comment #9: The terms "correct" or "appropriate" should be used uniformly and not interchangeably.

Response #9: We have changed the terminology to adequate and inadequate antibiotic therapy as per reviewer 1’s suggestion. We have made sure that this terminology is uniform throughout the manuscript. 

Comment #10: Can the authors describe what were the antibiotics that were provided inappropriately? for example, how many ESBL pathogens were covered by piperacillin-tazobactam and thus defined inappropriately treated (how many were susceptible to piperacillin-tazobactam)? how many SPICE pathogens were empirically covered by third generation cephalosporins and defined as inappropriate empiric therapy (how many were susceptible to third generation cephalosporins)?

Response #10: We have added this description (ESBL organisms treated with Piperacillin-Tazobactam, ESBL organisms susceptibility to Piperacillin-Tazobactam, SPICE organisms treated with Ceftriaxone, SPICE organism susceptibility to Ceftriaxone ) as per the reviewer’s suggestion (please see Results, sub-heading Empiric antibiotic choice, 4th paragraph, 2nd to 5th sentence). 

Comment #11: Table 4 – remove ER

Response #11: We have removed ER from Table 4 (please see Table 4). 

Comment #12: Page 14, line 216 – the fact that blood cultures were not drawn from the intravascular catheter does not preclude it from being the source of infection. I would not categorized it as "unknown source".

Response #13: Whether blood cultures were drawn from the intravascular catheter or not at the time of starting empiric antibiotics have nothing to do with classification of unknown source. The unknown source is based on the documentation of the clinician taking care of the patient when the clinician first started the empiric antibiotics. If the clinician did not find an infectious source and did not mention intravascular catheter as a possible source of infection, then it was considered to be an unknown source. We have tried to clarify the definition of suspected source in the manuscript (please see Methods, sub-heading Data collection, 1st paragraph, 3rd sentence). 

Comment #14: Table 2: please describe "other sources" in the footnotes.

Response #14: We have described the other sources in the footnote (please see Table 2 footnotes). 

Comment #15: Can the authors describe what were the cases where carbapenem was used un-necessarily? What was the antibiogram of Pseudomonas bacteremia? how many of these BSI were susceptible to each antibiotics?

Response #15: We described the cases where carbapenems were used unnecessarily (please see S1 Table 2). We also described the susceptibility profile of all the Pseudomonas isolates (please see S1 Table 1) as per the reviewer’s request. 

Comment #16: In the discussion would address the fact that guidelines were applicable only for ~40% of the cases.

Response #16: In the Discussion section 5th paragraph of the original manuscript, we wrote, “Second, although clinicians follow hospital guidelines for empiric antibiotic therapy in the majority of cases, these guidelines were only applicable in approximately half of bacteremia cases and a significant proportion of patients had an unknown source at initiation of empiric antibiotics. Therefore, a guideline on empiric antibiotic therapy for patients with sepsis and suspected bacteremia without a clear source may be helpful to guide clinicians.” We thought this was adequate to address the applicability of the guidelines. 

Comment #17: I believe it would be helpful to the readers if the authors could demonstrate how the NLR can be applied by showing pre-test and post-test probabilities for each risk factor and associated PLR and NLR.

Response #17: We have added a figure to show how the NLR will change from the pre-test to the post-test probability for ruling out Pseudomonas bacteremia using the clinical pathway of community associated infection as low risk versus clinician’s decision (please see S2 Supplemental Figure 1).

---

## [Decision Letter · Decision Letter 1]

8 Mar 2021

Local audit of empiric antibiotic therapy in bacteremia: A retrospective cohort study

PONE-D-20-40076R1

Dear Dr. Bai,

We’re pleased to inform you that your manuscript has been judged scientifically suitable for publication and will be formally accepted for publication once it meets all outstanding technical requirements.

Kind regards,

Dafna Yahav

Academic Editor

PLOS ONE

Additional Editor Comments (optional):

Reviewers' comments:

Reviewer's Responses to Questions

**Comments to the Author**

1. If the authors have adequately addressed your comments raised in a previous round of review and you feel that this manuscript is now acceptable for publication, you may indicate that here to bypass the “Comments to the Author” section, enter your conflict of interest statement in the “Confidential to Editor” section, and submit your "Accept" recommendation.

Reviewer #1: All comments have been addressed

Reviewer #2: All comments have been addressed

2. Is the manuscript technically sound, and do the data support the conclusions?

Reviewer #1: Yes

Reviewer #2: Yes

3. Has the statistical analysis been performed appropriately and rigorously? 

Reviewer #1: Yes

Reviewer #2: Yes

4. Have the authors made all data underlying the findings in their manuscript fully available?

Reviewer #1: Yes

Reviewer #2: Yes

5. Is the manuscript presented in an intelligible fashion and written in standard English?

Reviewer #1: Yes

Reviewer #2: Yes

6. Review Comments to the Author

Reviewer #1: Thank you for making these modifications. I have no further suggestions/comments. Good luck with dissemination of this important work.

Reviewer #2: I thank the authors for their thorough revision and edits. All my comments were responded appropriately.

I have not other comments.

7. PLOS authors have the option to publish the peer review history of their article (what does this mean?). If published, this will include your full peer review and any attached files.

Reviewer #1: **Yes: **Bradley Langford

Reviewer #2: No

---

## [Editor Report · Acceptance letter]

10 Mar 2021

PONE-D-20-40076R1 

Local audit of empiric antibiotic therapy in bacteremia: A retrospective cohort study 

Dear Dr. Bai:

I'm pleased to inform you that your manuscript has been deemed suitable for publication in PLOS ONE. Congratulations! Your manuscript is now with our production department. 

Kind regards, 

on behalf of

Dr. Dafna Yahav 

Academic Editor

PLOS ONE